# Transcriptomic Responses to Polymyxin B and Analogues in Human Kidney Tubular Cells

**DOI:** 10.3390/antibiotics12020415

**Published:** 2023-02-20

**Authors:** Mengyao Li, Mohammad A. K. Azad, Philip E. Thompson, Kade D. Roberts, Tony Velkov, Yan Zhu, Jian Li

**Affiliations:** 1Monash Biomedicine Discovery Institute, Infection Program and Department of Microbiology, Monash University, Melbourne, VIC 3800, Australia; 2Medicinal Chemistry, Monash Institute of Pharmaceutical Sciences, Monash University, Parkville, VIC 3052, Australia

**Keywords:** polymyxin, nephrotoxicity, transcriptomics, metallothioneins

## Abstract

Polymyxins are last-line antibiotics for the treatment of Gram-negative ‘superbugs’. However, nephrotoxicity can occur in up to 60% of patients administered intravenous polymyxins. The mechanisms underpinning nephrotoxicity remain unclear. To understand polymyxin-induced nephrotoxicity, human renal proximal tubule cells were treated for 24 h with 0.1 mM polymyxin B or two new analogues, FADDI-251 or FADDI-287. Transcriptomic analysis was performed, and differentially expressed genes (DEGs) were identified using ANOVA (FDR < 0.2). Cell viability following treatment with polymyxin B, FADDI-251 or FADDI-287 was 66.0 ± 5.33%, 89.3 ± 3.96% and 90.4 ± 1.18%, respectively. Transcriptomics identified 430, 193 and 150 DEGs with polymyxin B, FADDI-251 and FADDI-287, respectively. Genes involved with metallothioneins and Toll-like receptor pathways were significantly perturbed by all polymyxins. Only polymyxin B induced perturbations in signal transduction, including FGFR2 and MAPK signaling. SIGNOR network analysis showed all treatments affected essential regulators in the immune system, autophagy, cell cycle, oxidative stress and apoptosis. All polymyxins caused significant perturbations of metal homeostasis and TLR signaling, while polymyxin B caused the most dramatic perturbations of the transcriptome. This study reveals the impact of polymyxin structure modifications on transcriptomic responses in human renal tubular cells and provides important information for designing safer new-generation polymyxins.

## 1. Introduction

The polymyxins, namely polymyxin B and colistin (polymyxin E), have been used clinically since the late 1950s to treat infections caused by multidrug-resistant (MDR) Gram-negative bacteria, but were largely abandoned in the 1970s due to a high incidence of nephrotoxicity following intravenous administration [1,2]. However, the emergence of MDR Gram-negative bacteria resistant to virtually all currently available antibiotics has seen the reintroduction of the polymyxins into the clinic, in particular for treatment of *Acinetobacter baumannii*, *Pseudomonas aeruginosa* and *Enterobacterales* infections [3]. Given that the polymyxins retain substantial activity against many MDR Gram-negative bacteria, they are now often used as a last-line therapy to combat infections caused by these pathogens [4]. Nevertheless, nephrotoxicity remains the major dose-limiting factor for polymyxin use, occurring in up to 60% of patients [4]. Unfortunately, there is uncertainty regarding the mechanism by which polymyxins induce nephrotoxicity. Polymyxins are known to be extensively reabsorbed by and accumulate within kidney tubular cells [5,6], with necrosis subsequently occurring in a dose- and time-dependent manner [7]. Both polymyxin B and colistin have been shown to cause mitochondrial dysfunction and oxidative stress, which can trigger apoptosis [8,9]. Moreover, autophagy and polymyxin-induced tubular apoptosis involving the activation of death receptors, mitochondrial damage and endoplasmic reticulum stress have also been observed [7,8,9].

The key structural features of the polymyxins are the cationic γ-diaminobutyric acid (Dab) residues, the hydrophobic d-Phe^6^, l-Leu^7^ (polymyxin B) or *D*-Leu^6^ L-Leu^7^ (colistin) residues of the cyclic heptapeptide ring and the hydrophobic *N*-terminal fatty acyl group linked to the ring by a linear tripeptide side chain [10,11]. The antibacterial activity and toxicity of polymyxins are closely related to their structure [11]. Polymyxin analogues have previously been synthesized in an attempt to obtain high antimicrobial activity and low toxicity [12]. A group of new analogues were designed in our laboratory based on the polymyxin scaffold, with the aim of disconnecting therapeutic efficacy from toxicity [13]. To further investigate the impact of structural modifications on polymyxin nephrotoxicity, transcriptomics was employed to examine the effects of polymyxin B and two new analogues developed in our laboratory (FADDI-251 and FADDI-287) on human kidney proximal tubular HK-2 cells [14].

## 2. Results

### 2.1. Differentially Expressed Genes (DEGs) Shared by Polymyxin B, FADDI-251 and FADDI-287

Firstly, the viability of HK-2 cells following treatment with either 0.1 mM polymyxin B, FADDI-251 or FADDI-287 (Figure 1) was 66.0 ± 5.33%, 89.3 ± 3.96% and 90.4 ± 1.18% (mean ± standard deviation [15]), respectively, while it was 95.2 ± 2.11% for untreated control cells (Figure 2). Therefore, 0.1 mM was employed in the transcriptomics experiments.

Bioinformatic analysis revealed 432 DEGs associated with the treatment with polymyxin B (177 upregulated and 255 downregulated), 193 with FADDI-251 (95 upregulated and 98 downregulated) and 150 with FADDI-287 (81 upregulated and 69 downregulated) (Figure 3). All treatments resulted in the differential expression of a common set of 65 genes, with 58 genes (32 upregulated and 26 downregulated) consistently changed across the three treatments and seven inconsistently changed (e.g., upregulated by one treatment and downregulated by the remaining treatments). Function analysis on the 58 consistent DEGs was conducted by Reactome [16], with the top 10 pathways sorted by *p*-values listed in Table 1. The top three pathways identified, namely metallothioneins bind metals, response to metal ions and cellular responses to external stimuli, were mainly enriched by the metallothionein-encoding genes *MT1A, MT1B, MT1E, MT1F, MT1G, MT1H, MT1M* and *MT1X*. The only downregulated pathway, cellular responses to external stimuli, included the DEGs chromobox 8 (*CBX8*), nucleoporin 85 (*NUP85*) and inhibitor of DNA binding 1 (*ID1*). Genes *MT1M, MT1L*, inhibitor of nuclear factor kappa B kinase subunit beta (*IKBKB*) and insulin receptor substrate 1 (*IRS1*) were involved in Toll-like receptor (TLR) 1, TLR2 and TLR4 cascades. The citric acid (TCA) cycle and respiratory electron transport were also disturbed by significant upregulation of *COQ10B, LHX6, ND2* and *DLAT*, and downregulation of *NDUFS2*. Five of the inconsistently changed DEGs (*CHST3, WDR43, PREB, MB21D2* and *SLC38A5*) were upregulated by polymyxin B and downregulated by FADDI-251 and FADDI-287. In contrast, the remaining two inconsistently changed DEGs (diazepam binding inhibitor (*DBI)* and CXXC finger protein 4 (*CXXC4)*) were downregulated by polymyxin B and upregulated by FADDI-251 and FADDI-287 (Figure 4).

### 2.2. Polymyxin B Uniquely Induced Signaling Transduction by FGFR2 and MAPK

There were 310 unique genes differentially expressed by polymyxin B treatment (Figure 3). Of these genes, those involved in fibroblast growth factor receptor 2 (*FGFR2*) and mitogen-activated protein kinase (*MAPK*)-signaling transduction pathways were significantly over-represented (false discovery rate (FDR) < 0.2) (Table 2). In the FGFR2 pathway, dual specificity phosphatase 2 (*DUSP2*) was upregulated, whereas chromosome 11 open reading frame 54 (*C11orf54*) and fibroblast growth factor receptor 2 (*FGFR2*) were downregulated (Table 2). Significant enrichment in the MAPK-mediated signaling pathways included MAPK family signaling cascades and the RAF/MAP kinase cascade (Table 2). Upregulated genes were also represented in the MAPK family signaling cascades, including *DUSP2*, major facilitator superfamily domain containing 1 (*MFSD1*), p21 (*RAC1*) activated kinase 2 (*PAK2*), C-X-C motif chemokine ligand 2 (*CXCL2*) and Jun proto-oncogene (*JUN*). With the exception of *C11orf54* and *FGFR2*, the MAPK family signaling cascades comprised downregulated genes such as discs large MAGUK scaffold protein 3 (*DLG3*), proline rich nuclear receptor coactivator 1 (*PNRC1*), solute carrier family 37 member 4 (*SLC37A4*), 6-phosphofructo-2-kinase/fructose-2,6-biphosphatase 4 (*PFKFB4*), persephin (*PSPN*) and E74-like ETS transcription factor 3 (*ELF3*). The RAF/MAP kinase cascade pathway was over-represented with the same downregulated genes as the MAPK family signaling cascades. The 29 and 25 unique DEGs induced by FADDI-251 and FADDI-287, respectively, were not enriched in any pathway (FDR < 0.2).

### 2.3. Networks Perturbed by Polymyxin B and Analogues

The networks perturbed by all three treatments (polymyxin B, FADDI-251 and FADDI-287) were retrieved from the SIGnaling Network Open Resource (SIGNOR) database [17] of causal relationships between biological entities (Figure 5A). There were nine genes significantly dysregulated by all the three treatments (Appendix A). *IKBKB* (degree = 31) and *IRS1* (degree = 30) were the top two hub genes by degree. *IKBKB* and its neighbors in the protein–protein interaction (PPI) network (e.g., *FOXO, RUNX3, CASP3, MAP3K7* and *MAP3K14*) play essential roles in immune responses, autophagy, the cell cycle, oxidative stress-induced senescence, gene expression, protein metabolism, apoptosis and signal transduction. *IRS1* and its neighbors (including *PIK3CA, MAPK1, MAPK8, MAPK9, MAP2K1, TNF* and *PLK1*) are involved in the cell cycle, immune responses, cellular senescence, gene expression, apoptosis and signal transduction.

Nine DEGs were induced only by polymyxin B with a degree greater than 10 in the PPI network, showing the exclusive alterations caused by polymyxin B. *JUN* (degree = 51) and its neighbors are involved in interleukin signaling in the immune system and RNA polymerase II transcription. *STAT1* (degree = 38) and its neighbors are involved with cytokine signaling, including interferon and interleukin signaling. *EGR1* (degree = 28), *BDKRB2* (degree = 15) and *FGFR2* (degree = 13) play essential roles in signal transduction, whereas *MAP2K6* (degree = 14) and *SH3RF1* (degree = 11) are associated with the immune system. The remaining two polymyxin B-specific hub genes were *CEBPA* (degree = 28), a transcription factor for genes regulating the cell cycle and *CREB5* (degree = 12), which encodes a homodimer binding to the cAMP response element. There were no hub genes with a degree greater than 10 induced by FADDI-251 or FADDI-287. Collectively, all three polymyxins affected cellular immune responses, the cell cycle, apoptosis and gene expression, although polymyxin B induced substantially more DEGs in these pathways.

## 3. Discussion

Nephrotoxicity is the major dose-limiting factor associated with intravenously administered polymyxins, occurring in up to 60% of patients [4]. New polymyxin analogues have been developed with the hope of increasing antibacterial activity whilst reducing nephrotoxicity [18]. Pharmacokinetic and in vitro studies have demonstrated the extraordinary accumulation of polymyxin B and colistin in the kidneys, in particular kidney tubular cells [19,20]. In this chemical biology study, we conducted transcriptomics to compare the cellular responses of human kidney proximal tubular HK-2 cells to polymyxin B and two analogues, FADDI-251 and FADDI-287. Compared to polymyxin B, the structural differences in analogues FADDI-251 and FADDI-287 involve changes at positions three (where l-Dap replaces l-Dab with one-carbon less in the side chain), six (where d-Leu replaces d-Phe) and seven (where l-Thr in FADDI-251 and l-Abu in FADDI-287 replace l-Leu). The only difference between the structures of FADDI-251 and FADDI-287 is a single hydroxyl group at position seven. These modifications maintain the key structural features of the polymyxin core but do decrease the overall hydrophobicity of the polymyxin core structure. Meanwhile, these modifications do help attenuate nephrotoxicity [14] and are critical for the interactions of polymyxins with the kidney cell membrane [21]; thus, these less nephrotoxic polymyxin analogues, FADDI-251 and FADDI-287, serve as useful tools for exploring the transcriptomic impact of the polymyxins on kidney cells to further our understanding of the mechanism of polymyxin induced-nephrotoxicity. Importantly, improved antibacterial activities of FADDI-251 and FADDI-287 have been reported against Gram-negative bacterial isolates [13]. Our current study demonstrated that, by rationally modifying multiple non-conserved positions in the polymyxin scaffold, the nephrotoxicity of new polymyxin molecules such as FADDI-287 can be significantly decreased. Importantly, a recent mouse nephrotoxicity study confirmed the reduced nephrotoxicity of FADDI-287 compared to polymyxin B in vivo [13].

According to the previously reported in vitro studies in HK-2 cells (EC_50_ approximately 0.35 mM), 0.1 mM polymyxin B was employed to avoid substantial cell death and to maintain the quality of mRNA collected for subsequent transcriptomics analysis [9,22]. Cell death caused by the two polymyxin analogues was significantly lower than that with polymyxin B. A major finding of our study is that metallothionein genes and the TLR cascades were significantly perturbed by all the three polymyxins, indicating that metal homeostasis and the immune system were essential host responses to polymyxin treatment. Furthermore, vital regulators involved with immune responses, autophagy, the cell cycle, oxidative stress-induced senescence, gene expression, protein metabolism, apoptosis and signal transduction were also perturbed by all three polymyxins. Unlike FADDI-251 and FADDI-287, polymyxin B caused the largest number of DEGs, including those involved in FGFR2 and MAPK signaling transduction. Our results confirmed that hydrophobicity of position six of the polymyxin core structure is critical in the nephrotoxicity.

Metallothioneins are a family of small, cysteine-rich heavy metal-binding proteins involved in essential trace element homeostasis and metal detoxification [23]. With metal-binding and redox capabilities, metallothioneins can protect cells against metal toxicity and oxidative stress [24,25]. All three treatments resulted in the transcriptional upregulation of a large number of genes from the metallothionein family (*MT1A, MT1B, MT1E, MT1F, MT1G, MT1H, MT1M, MT1X* and pseudogene *MT1L*), suggesting that each induced oxidative stress. Our group and collaborators have previously reported that oxidative stress is a key factor in colistin-induced nephrotoxicity [7,26]. Dai et al. [7] reported that an oxidative stress marker, malondialdehyde, was significantly increased, and two key antioxidant enzymes (superoxide dismutase (SOD) and catalase) were significantly decreased in mouse kidneys following 7 days of colistin treatment (7.5 and 15 mg/kg/day). These results indicated a decreased ability of kidney HK-2 cells to catalyze the decomposition of reactive oxygen species. In the present transcriptomics study, we identified the metallothionein-encoding genes as potential indicators of polymyxin-induced oxidative stress.

There were seven inconsistent DEGs induced by polymyxin B and the two analogues, of which *SLC38A2* and *CXXC4* sparked our interest. *SLC38A2* encodes the sodium-dependent neutral amino acid transporter 2 (*SNAT2*), and it is upregulated by amino acid starvation and hypertonicity [27]. *SLC38A2* was significantly upregulated by polymyxin B but downregulated by FADDI-251 or FADDI-287, suggesting that polymyxin B caused nutrient stress via amino acid starvation in HK-2 cells. The CXXC4 protein is a negative regulator of Wnt and Ras/MAPK signaling, with the downregulation of CXXC4 promoting malignant phenotype in renal cell carcinoma by activating Wnt signaling [28,29]. As a tumor suppressor, CXXC4 inhibits cell growth by activating apoptosis in gastric cancer [30]. Interestingly, *CXXC4* was downregulated by polymyxin B but upregulated by FADDI-251 or FADDI-287, indicating that apoptosis could be induced by polymyxin B. Furthermore, on the one hand, CXXC4 could be used as an indicator for polymyxin toxicity, while on the other hand, the less toxic polymyxin analogues could be considered anti-tumor drugs by targeting CXXC4.

Another major finding of our study is that polymyxin B exclusively perturbed the signaling pathways FGFR2 and MAPK, suggesting multiple signaling transduction pathways contribute to polymyxin-induced nephrotoxicity. The fibroblast growth factor (FGF) family activates FGF receptors (FGFRs) to induce pleiotropic responses that control cell proliferation, differentiation, migration, survival and shape in a context-dependent manner [31]. As receptor tyrosine kinases, FGFR1-FGFR4 are composed of an extracellular immunoglobulin-like domain, a single transmembrane domain and a cytoplasmic domain containing the catalytic protein tyrosine kinase core and regulatory sequences [32,33]. Among the FGFRs, *FGFR2*, which transduces FGF signals to RAS-MAPK and PI3K-AKT signaling cascades [31,34], was significantly decreased by polymyxin B. MAPK and PI3K—the AKT pathways—are fundamental signal transduction and regulation pathways involved in the majority of cellular physiological processes. Activation of these pathways is responsible for cell proliferation, differentiation, metabolism, cytoskeleton reorganization, death and survival [35,36,37,38]. Both MAPK and PI3K-AKT pathways have previously been shown to be involved in colistin-induced nephrotoxicity [7] and neurotoxicity [39], suggesting that these pathways and their downstream cell death are hallmarks of polymyxin-induced toxicity. In the present study, genes from the MAPK and PI3K-AKT pathways were only dysregulated by polymyxin B, indicating that its greater toxicity could be attributed to activation of the intracellular signaling cascades. Furthermore, the potential for reduced toxicity with FADDI-251 (hub DEGs = 14) and FADDI-287 (hub DEGs = 7) was also shown by the PPI networks where polymyxin B (hub DEGs = 26) perturbed more differentially expressed hub genes than the two analogues, including *JUN,* which is involved in the MAPK signaling cascade. Nine hub genes shared by all three polymyxins play crucial regulatory roles in immune responses, cell cycle, cell death, oxidative stress and signal transduction, indicating the possibility that polymyxin B and analogues all caused substantial disturbances in these functions [40]. All these responses could be associated with the physical binding of DNA and polymyxins, which directly leads to DNA damage and oxidative stress [40] and subsequent biological events, including immune responses and cell cycle arrest.

## 4. Materials and Methods

### 4.1. Polymyxin B and Polymyxin Analogues

Polymyxin B (Batch number 20120204) was obtained from Beta Pharma (Shanghai, China). Polymyxin analogues FADDI-251 and FADDI-287 (Figure 1) were synthesized using standard Fmoc solid-phase peptide synthesis techniques, which have previously been described elsewhere [10,19].

### 4.2. Assessment of Cell Viability

Human kidney proximal tubule HK-2 cells (ATCC CRL-2190™) were initially grown in keratinocyte serum-free medium (KSFM) supplemented with pituitary growth hormone (Life Technologies, Thermo Fisher Scientific, Scoresby, VIC, Australia) at 37 °C in a humidified atmosphere containing 5% CO_2_. For assessment of cell viability, cells were seeded and grown for 24 h in 24-well plates (0.5 × 10^5^ cells/well, 1.0 mL) before treatment with 1.0 mL of 0.1 mM polymyxin B, FADDI-251 or FADDI-287 for 24 h. Stock solutions of polymyxin B or its analogues (10.0 mM) were prepared in phosphate-buffer saline (PBS; Gibco, Life Technologies, Thermo Fisher Scientific, Scoresby, VIC, Australia) and sterilized by passage through a 0.20-μm cellulose acetate syringe filter (Millipore, Bedford, MA, USA) immediately prior to all experiments. Control cells were treated with the vehicle (10.0 µL/mL of PBS) for 24 h. Following treatment, cells were detached using trypsin-EDTA solution (0.05%; Gibco, Life Technologies, Thermo Fisher Scientific, Scoresby, VIC, Australia), stained with propidium iodide (PI, 1.5 µM, red fluorescence, Ex/Em = 493/535–617 nm; Invitrogen, Thermo Fisher Scientific, Scoresby, VIC, Australia) and their viability was measured using flow cytometry (Novocyte Flow Cytometer; ACEA Biosciences, Inc., San Diego, CA, USA) [41].

### 4.3. Extraction of mRNA

Cells were seeded and grown for 24 h in 6-well plates (2.5 × 10^5^ cells/well, 3.0 mL) using the same growth media described above for cell viability assessments. Following treatment with 0.1 mM polymyxin B or analogues (or no treatment for controls), the growth medium was aspirated completely, and lysis buffer added to the cell-culture dish. A rubber policeman was used to detach the cells. The lysate was passed through a blunt 20-gauge needle fitted to an RNase-free syringe at least 5 times before processing according to the RNeasy Plus Mini Kit (QIAGEN, Clayton, VIC, Australia). 

Quantification of the extracted mRNA was performed by the Monash Health Translation Precinct (MHTP) Medical Genomic Facility with Agilent microarray using the Human Gene Expression v2. In brief, Cyanine-3 (Cy3) labelled cRNA was prepared using the One-Color Low input Quick Amp labelling Kit (Agilent, Mulgrave, VIC, Australia), followed by RNeasy column purification as described above. Dye incorporation and cRNA yield were checked with a NanoDrop ND-1000 Spectrophotometer. The Cy3 labeled cRNA was fragmented and hybridized with the Human Gene Expression v2 at 65 °C for 17 h. After washing, the slides were scanned with a DNA microarray scanner using one-color scan settings for 8x60k array slides (GeneSpring, GX11.5; Agilent). The scanned images were analyzed with Feature Extraction Software 11.0.1.1 (Agilent) using default parameters.

### 4.4. Bioinformatic Analysis

The raw intensities were pre-processed to perform background correction, quantile normalization and log_2_ transformation by Bioconductor package Limma [42]. If multiple probes were mapped to the same gene, the expression value for the gene was summarized as the arithmetic mean of the values of the multiple probes (on the log_2_ scale). DEGs were identified using analysis of variance (ANOVA) with an FDR < 0.2. All *p*-values in this paper were adjusted by the Benjamini–Hochberg FDR procedure [43]. The SIGNOR database [17], which has collected approximately 12,000 causal relationships between over 2800 human proteins, was employed to construct the directed gene network. Networks were visualized by Cytoscape [44]. The functional pathway analysis was performed against Reactome hierarchy [16]. The hypergeometric distribution was used to determine whether a pathway was over-represented in the interesting gene list (i.e., DEG list).

## 5. Conclusions

This study revealed metal homeostasis by metallothioneins and Toll-like receptor cascades as major cellular responses involved in polymyxin-induced toxicity. Only polymyxin B activated the FGFR2 and MAPK signaling pathways, indicating that decreasing the hydrophobicity of the polymyxin scaffold can substantially attenuate polymyxin-induced perturbations in kidney tubular cells and the nephrotoxicity. Our structure–toxicity relationship findings may facilitate the discovery of new-generation, safer polymyxins.

## Figures and Tables

**Figure 1 antibiotics-12-00415-f001:**
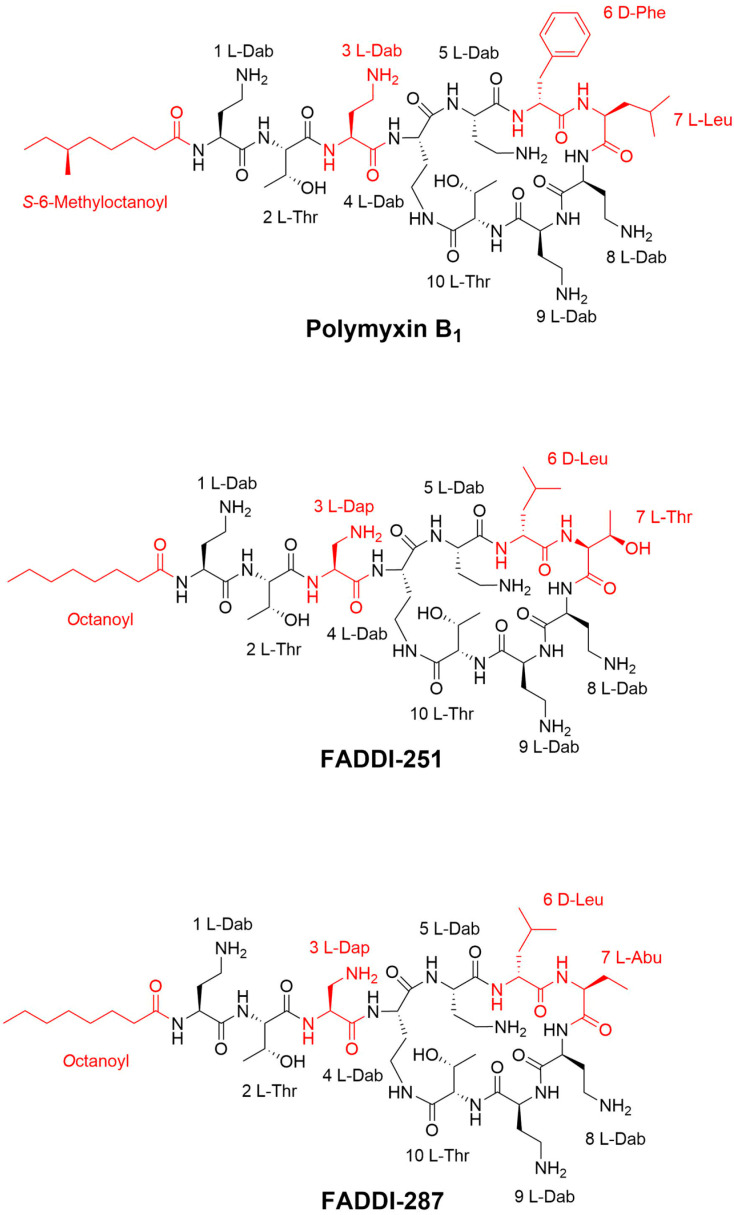
Chemical structures of polymyxin B_1_, FADDI-251 and FADDI-287.

**Figure 2 antibiotics-12-00415-f002:**
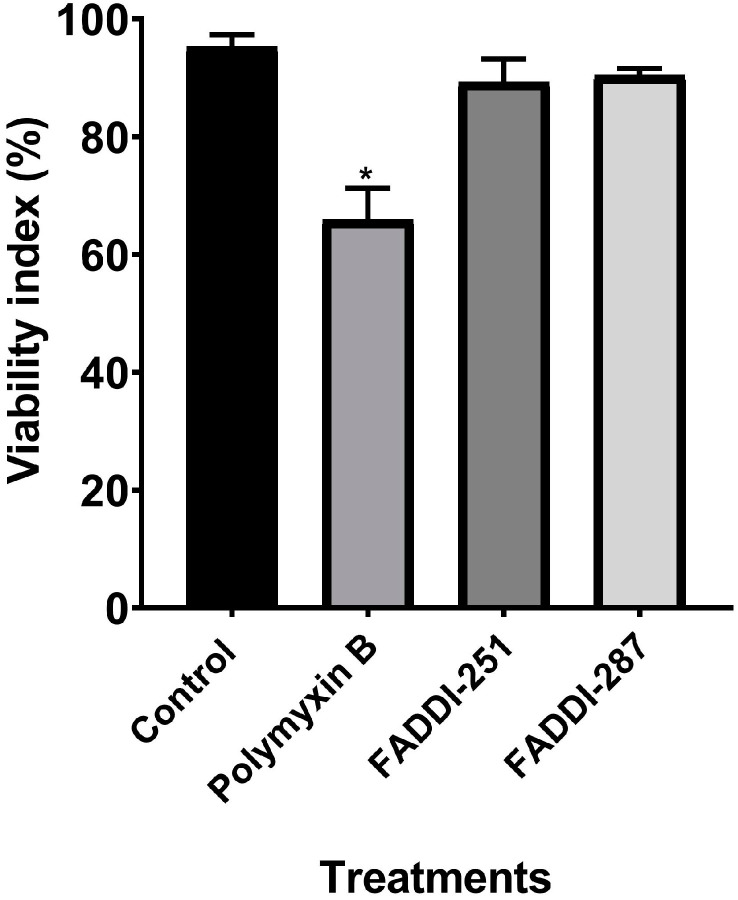
Cell viability of HK-2 cells following a 24-h treatment with 0.1 mM polymyxin B, FADDI-251 or FADDI-287, and the control without treatment. Significance: * *p* ≤ 0.0001; multiple comparison Tukey test.

**Figure 3 antibiotics-12-00415-f003:**
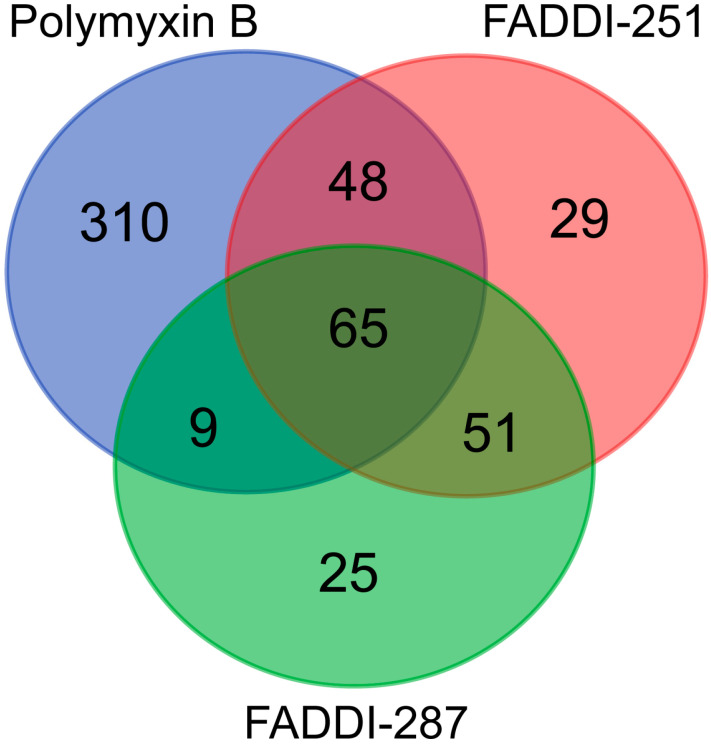
Numbers of common and unique DEGs in HK-2 cells induced by polymyxin B, FADDI-251 and FADDI-287.

**Figure 4 antibiotics-12-00415-f004:**
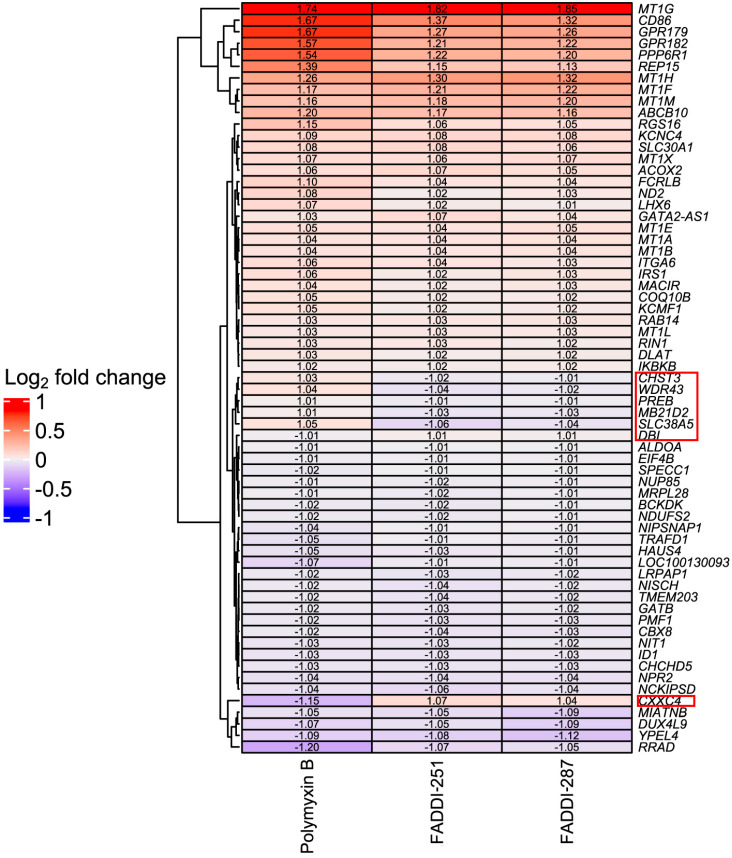
Heatmap of common DEGs and Log_2_ fold changes in HK-2 cells induced by polymyxin B, FADDI-251 and FADDI-287. Inconsistently changed genes across three treatments are indicated by the red boxes on the right.

**Figure 5 antibiotics-12-00415-f005:**
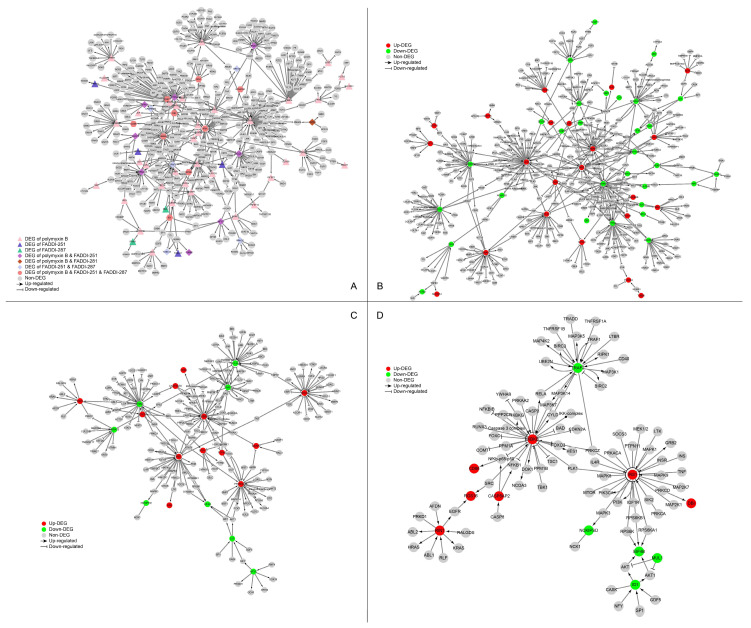
Perturbation of the signaling network of HK-2 cells by polymyxin B, FADDI-251 and FADDI-287. (**A**) Perturbation of the signaling network by any polymyxin. Triangle nodes represent the DEGs induced by only one polymyxin, diamond nodes represent the DEGs induced by two polymyxins and hexagon nodes represent the DEGs induced by all the three polymyxins. (**B**–**D**) Perturbation of the signaling network by polymyxin B, FADDI-251 and FADDI-287, respectively. Red nodes represent the upregulated DEGs induced by polymyxin, and green nodes represent the downregulated DEGs induced by polymyxin. Enlarged for details.

**Table 1 antibiotics-12-00415-t001:** Top 10 pathways enriched by 58 DEGs common to treatments by polymyxin B, FADDI-251 and FADDI-287.

Pathway	*p*-Value	FDR	Upregulated	Downregulated
Metallothioneins bind metals	0.28 × 10^−15^	0.14 × 10^−12^	*MT1A, MT1B, MT1E, MT1F, MT1G, MT1H, MT1M, MT1X*	_
Response to metal ions	0.24 × 10^−14^	0.58 × 10^−12^	*MT1A, MT1B, MT1E, MT1F, MT1G, MT1H, MT1M, MT1X*	_
Cellular responses to external stimuli	0.8 × 10^−5^	0.13	*MT1A, MT1B, MT1E, MT1F, MT1G, MT1H, MT1M, MT1X*	*CBX8, NUP85, ID1*
Toll-like Receptor TLR1:TLR2 Cascade	0.27 × 10^−2^	0.18	*MT1M, MT1L, IKBKB, IRS1*	_
Toll-like Receptor 2 (TLR2) Cascade	0.27 × 10^−2^	0.18	*MT1M, MT1L, IKBKB, IRS1*	_
PIPs transport between Golgi and plasma membranes	0.5 × 10^−2^	0.18	*MT1H*	_
Signaling by NTRK1 (TRKA)	0.6 × 10^−2^	0.18	*MT1M, IRS1*	*ID1, RRAD*
Toll-like Receptor 4 (TLR4) Cascade	0.61 × 10^−2^	0.18	*MT1M, MT1L, IKBKB, IRS1*	_
The citric acid (TCA) cycle and respiratory electron transport	0.65 × 10^−2^	0.18	*COQ10B, LHX6, ND2, DLAT*	*NDUFS2*
Nuclear events (kinase and transcription factor activation)	0.78 × 10^−2^	0.18	*MT1M*	*ID1, RRAD*

**Table 2 antibiotics-12-00415-t002:** Reactome pathways enriched by unique DEGs induced by polymyxin B only.

Pathway	*p*-Value	FDR	Upregulated	Downregulated
SHC-mediated cascade: FGFR2	3.62 × 10^−5^	0.03	*DUSP2*	*C11orf54, FGFR2*
Signaling by FGFR2 IIIa TM	5.36 × 10^−5^	0.03		*C11orf54, FGFR2*
Downstream signaling of activated FGFR2	1.6 × 10^−4^	0.06	*DUSP2*	*C11orf54, FGFR2*
FGFR2 mutant receptor activation	4.01 × 10^−4^	0.11		*C11orf54, FGFR2*
Phospholipase C-mediated cascade; FGFR2	6.2 × 10^−4^	0.11		*C11orf54, FGFR2*
FGFR2 ligand binding and activation	7.38 × 10^−4^	0.11		*C11orf54, FGFR2*
IGF1R signaling cascade	8.13 × 10^−4^	0.11	*DUSP2*	*C11orf54, FGFR2*
Signaling by Type 1 Insulin-like Growth Factor 1 Receptor (IGF1R)	8.87 × 10^−4^	0.11	*DUSP2*	*C11orf54, FGFR2*
PI-3K cascade: FGFR2	0.16 × 10^−2^	0.18		*C11orf54, FGFR2*
Signaling by FGFR2 in disease	0.19 × 10^−2^	0.19		*C11orf54, FGFR2*
FRS-mediated FGFR2 signaling	0.24 × 10^−2^	0.19		*C11orf54, FGFR2*
MAPK family signaling cascades	0.13 × 10^−2^	0.16	*DUSP2, CXCL2, JUN, MFSD1, PAK2*	*C11orf54, FGFR2, DLG3, PNRC1, SLC37A4, PFKFB4, PSPN, ELF3*
RAF/MAP kinase cascade	0.24 × 10^−2^	0.19	*DUSP2, CXCL2*	*C11orf54, FGFR2, DLG3, PNRC1, SLC37A4, PFKFB4, PSPN, ELF3*

## Data Availability

The microarray data are available from the authors upon reasonable request.

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
