# Peer review of "Transcriptomic Responses to Polymyxin B and Analogues in Human Kidney Tubular Cells"

_antibiotics, 2023, doi:10.3390/antibiotics12020415_

Round 1

Reviewer 1 Report

more illustration in tabular form on two analogues FADDI 251 & 287 in discussion will give more impact to readers.

Author Response

Reviewer #1:

More illustration in tabular form on two analogues FADDI 251 & 287 in discussion will give more impact to readers.

Response: We thank the reviewer for this comment. Initially, we had also considered presenting the genes or pathways perturbed by polymyxin B and the analogues in tabular form. However, we found that is not the most efficient way to show the differences between them. Therefore, we chose to use network visualization for both the DEGs and hub regulators altered by different analogues.

Reviewer 2 Report

Section

Comments and recommendation

General comment

The manuscript entitled: Transcriptomic responses to polymyxin B and analogues in hu-2 man kidney tubular cells, focuses on an important topic in pharmaceutical sciences and pharmacogenomics. Indeed, the idea of the manuscript is novel and it is well written using the standard journal format.

Abstract

-The abstract is well described and contained all required information of the study with clear objectives.

Introduction

-The introduction section is well written and described the well the state of the art of this work problem. In addition, the authors have used up to date research articles relevant to the study topic.

Materials and methods

-The design of this research work fits well with the aim and scope of this type of investigations. The methodologies used in are in accordance with type of work. In addition, the authors have used suitable statistical model to analyze this type of data.

Results

-The results of this manuscript are described the findings of the manuscript.

Discussion

-The results are well discussed in the discussion section using up to date literature which closely related to the study topic.

Conclusions

-The conclusions of this manuscript described well the manuscript findings.

Bibliography/References

-The list of references is well formatted according to journal instructions. Most of references listed in the manuscript is up to date and closely relative to the research focus of study.

Recommendation and final comment

-I recommend acceptance of this manuscript entitled: Transcriptomic responses to polymyxin B and analogues in hu-2 man kidney tubular cells.  The evaluation is based on that overall value of data presented and novelty of the idea. In addition, it is on the scope of this outstanding journal.

Author Response

Reviewer #2:

Recommendation and final comment: I recommend acceptance of this manuscript entitled: Transcriptomic responses to polymyxin B and analogues in human kidney tubular cells. The evaluation is based on that overall value of data presented and novelty of the idea. In addition, it is on the scope of this outstanding journal.

Response: We thank the reviewer for the recommendation.

Reviewer 3 Report

This manuscript present us that HK-2 cells response to polymyxin B, FADDI-251 and FADDI-287, which is of interest. I have two minor comments.

1.  In terms of this manuscript, FADDI-251 and FADDI-287 have lower nephrotoxicity. What are their bioactivity?

2. How dose the different signalling pathway of  polymyxin B, FADDI-251 and FADDI-287 affect the nephrotoxicity?

Author Response

Reviewer #3:

This manuscript presents us that HK-2 cells response to polymyxin B, FADDI-251 and FADDI-287, which is of interest. I have two minor comments.

  1. In terms of this manuscript, FADDI-251 and FADDI-287 have lower nephrotoxicity. What are their bioactivity?

Response: The antibacterial activity of FADDI-251 and FADDI-287 have been described in the Discussion section of the revised manuscript (lines 197-198).

  1. How does the different signalling pathway of  polymyxin B, FADDI-251 and FADDI-287 affect the nephrotoxicity?

Response: Compared to FADDI-251 and FADDI-287, polymyxin B induced perturbations in more pathways associated with cell death, such as MAPK signalling pathway.  The Discussion section has been modified accordingly (lines 213-214).

Reviewer 4 Report

Title: Transcriptomic responses to polymyxin B and analogues in human kidney tubular cells

This manuscript is well-written by the authors. I do believe that if they can improve the manuscripts following all comments. It might have a chance to publish in the journal.

Comments

Abstract

1. The first sentence of the abstract should be described the important of polymyxin B, followed by its effects on the Nephrotoxicity.

2. Please identify the objective of this study. It may be better if the author uses passive voice instead of, we propose ……

3. The word “Transcriptomics was”. It should be replaced by “Transcriptomic analysis was”

4. The overall of the abstract is well-written.

Introduction

5. Line 42-44, please add the use of polymyxins for the treatment of the bacterial infection.

6. Line 45-48, please add the example bacteria that's resistant to polymyxins.

7. Please identify or describe the information of polymyxin analogues FADDI-251 or FADDI-287 in the introduction.

8. Please include the objectives of the study. It is better if the authors use passive voice.

Methods

9. Please delete “To assess cell viability and mRNA extraction (discussed below),” .

10. Line 293-294, how long of the incubation time for the treatment?

Results

11. Line 72, please remove to the methods.

12. Figure 2, the authors should use only one star (*) for the significant difference. In the figure, the authors use four stars (****). Please include the significant difference in the statistical analysis.

13. Line 83, those metallothionein-encoding genes MT1A, MT1B, MT1E, MT1F, 83 MT1G, MT1H, MT1M and MT1X should be written in italic. All the genes should be written in italic.

14. Figure 5, the size of the letter is small.

15. The authors should summarize the up and down regulation of the genes due to the pathway. It is easy to understand, and it can describe the gene expression to predict the mechanisms of the drugs against the cells.   

Discussion

16. The authors shouldn’t repeat the results in the discussion.

17. Please remove the words (Figure 1, Figure 2…) from the discussion.

18. Please describe and discuss the compact and key the results.

References.

19. There are many references. In general, there are 30-40 references for each manuscript (research articles). Please delete some unnecessary references or old references.

Author Response

Reviewer #4:

This manuscript is well-written by the authors. I do believe that if they can improve the manuscripts following all comments. It might have a chance to publish in the journal.

Abstract

  1. The first sentence of the abstract should describe the importance of polymyxin B, followed by its effects on the nephrotoxicity.

Response: We thank the reviewer for the comment. The importance of polymyxin B has been included in the first sentence of the Abstract.

  1. Please identify the objective of this study. It may be better if the author uses passive voice instead of, we propose ……

Response: The objective of the study is to elucidate the mechanism underpinning polymyxin-associated nephrotoxicity; it is now clearly stated in the revised manuscript (lines 25-26).

  1. The word “Transcriptomics was”. It should be replaced by “Transcriptomic analysis was”

Response: The manuscript has been revised accordingly (line 27).

  1. The overall of the abstract is well-written.

Response: We thank the reviewer for their kind comment.

Introduction

  1. Line 42-44, please add the use of polymyxins for the treatment of the bacterial infection.

Response: The clinical use of polymyxins has been added (lines 44-45).

  1. Line 45-48, please add the example bacteria resistant to polymyxins.

Response: Examples of polymyxin-resistant bacteria have been provided in the revised manuscript (lines 48-50).

  1. Please identify or describe the information of polymyxin analogues FADDI-251 or FADDI-287 in the introduction.

Response:  FADDI-251 and FADDI-287 have now been mentioned in the Introduction (lines 68-70).

  1. Please include the objectives of the study. It is better if the authors use passive voice.

Response: The study objective has now been included (lines 70-73).

Methods

  1. Please delete “To assess cell viability and mRNA extraction (discussed below),”.

Response: This has been deleted.

  1. Line 293-294, how long of the incubation time for the treatment?

Response: The incubation time of 24 h has now been included in the revised manuscript (line 298).

Results

  1. Line 72, please remove to the methods.

Response: Cell viability was tested to confirm cell death after polymyxin treatment. Therefore, we consider it part of the results and have left this as it was (now line 76).

  1. Figure 2, the authors should use only one star (*) for the significant difference. In the figure, the authors use four stars (****). Please include the significant difference in the statistical analysis.

Response: The figure has been modified accordingly and the P value (P0.0001) has been included in the figure caption (lines 154-155).

  1. Line 83, those metallothionein-encoding genes MT1A, MT1B, MT1E, MT1F, 83 MT1G, MT1H, MT1M and MT1X should be written in italic. All the genes should be written in italic.

Response: All genes have been italicized throughout the manuscript.

  1. Figure 5, the size of the letter is small.

Response: We apologize for the small font size. However, due to the large number of nodes in these networks, if the font size is enlarged the gene names in each node will overlap with neighbours. Therefore, the resolution of this figure has been set to 300 dpi so that readers can zoom in to check gene names.

  1. The authors should summarize the up and down regulation of the genes due to the pathway. It is easy to understand, and it can describe the gene expression to predict the mechanisms of the drugs against the cells.  

Response: We thank the reviewer for this comment. The up- and down-regulation of genes has been presented as suggested in Tables 1 and 2.

Discussion

  1. The authors shouldn’t repeat the results in the discussion.

Response: We agree with the reviewer on this point and have mainly focused on the discussion.

  1. Please remove the words (Figure 1, Figure 2…) from the discussion.

Response: These words have been deleted as suggested.

  1. Please describe and discuss the compact and key the results.

Response: This transcriptomic study revealed the structure-toxicity relationship of polymyxin B and two structural analogues. Only polymyxin B activated the FGFR2 and MAPK signalling pathways, indicating that decreasing the hydrophobicity of the polymyxin scaffold can substantially attenuate polymyxin-induced perturbations in kidney tubular cells. This has been directly stated at the end of the Discussion (lines 275-281).

References.

  1. There are many references. In general, there are 30-40 references for each manuscript (research articles). Please delete some unnecessary references or old references.

Response: The number of references has been reduced accordingly.